# Mechanism of Antitumor Effects of Saffron in Human Prostate Cancer Cells

**DOI:** 10.3390/nu16010114

**Published:** 2023-12-28

**Authors:** Mohammad Khan, Kaitlyn Hearn, Christian Parry, Mudasir Rashid, Hassan Brim, Hassan Ashktorab, Bernard Kwabi-Addo

**Affiliations:** 1Department of Biochemistry and Molecular Biology, Howard University, Washington, DC 20059, USA; masud1234@yahoo.com; 2Department of Life Sciences, Xavier University of Louisiana, New Orleans, LA 70125, USA; kaitlynhearn5@gmail.com; 3Department of Microbiology, Howard University, Washington, DC 20059, USA; christian.parry@howard.edu; 4Cancer Center, Howard University, Washington, DC 20059, USA; mudasir.rashid@howard.edu (M.R.); hashktorab@howard.edu (H.A.); 5Department of Pathology, Howard University, Washington, DC 20059, USA; hbrim@howard.edu

**Keywords:** saffron, prostate cancer cell lines, DNA repair genes, epigenetics, anti-tumor

## Abstract

Prostate cancer is the most common cancer and the second leading cause of cancer deaths among men in the USA. Several studies have demonstrated the antitumor properties of saffron in different types of cancers, including prostate cancer. The oral administration of saffron extract has been reported to have antitumor effects on aggressive prostate-cancer-cell-line-derived xenografts in nude male mice. The objective of this study was to carry out in vitro studies of saffron-treated prostate cancer cells to ascertain the effects of saffron on key intermediates in prostate carcinogenesis. Our studies demonstrated the significant inhibition of cell proliferation for androgen-sensitive prostate cancer cell lines via apoptotic pathways. We also demonstrate the statistically significant down-regulation of DNA methyltransferases (COMT, MGMT, EHMT2, and SIRT1 deacetylase) in saffron-treated prostate cancer cells. In addition, saffron-treated prostate cancer cells displayed a statistically significant dysregulation of DNA repair intermediates (WRN, p53, RECQ5, MST1R, and WDR70) in a time-dependent manner. Furthermore, Western blot analysis demonstrated that saffron treatment induced changes in the expression of other key genes (DNMT1, DNMT3b, MBD2, CD44, HDAC3, c-Myc, NF-kB, TNFα, AR, N-RAS, and PTEN) in prostate cancer cells. Collectively, our findings demonstrate the important mechanisms by which saffron mediates anti-tumor properties in prostate cancer. These findings suggest that the use of saffron supplements alongside standard treatment protocols may yield beneficial effects for individuals with prostate cancer.

## 1. Introduction

The high incidence and mortality rates from prostate cancer (PCa) makes this disease a major public health concern. The American Cancer Society currently estimate that 268,490 men will be diagnosed with PCa and 34,500 men will die of their disease in the United States of America alone [1]. The well-established risk factors for PCa includes age, as the disease is commonly detected in men over the age of 55 years old; this peaks around the age of 65 years old, when approximately 80% of the disease cases are detected [2]. Prostate cancer incidence and mortality rates also present persistent racial/ethnic disparities among the USA population. African American men (AA, men of African ancestry) are disproportionately affected by PCa; AA men are generally diagnosed with PCa at a 1.7-fold higher rate and die from the disease at more than a 2-fold higher rate compared to their European American (EA) counterparts, who rank second in terms of the disease incidence and mortality rates [3] by race/ethnic category, with Asian and Pacific Islander men presenting the lowest incidence and mortality rates. Another important risk factor for PCa is family history: having a brother or father diagnosed with the disease increases your risk of PCa by 2- to 4-fold [4], suggesting biological factors, shared environmental factors, or familial predispositions to PCa. Biological factors may also account for a proportion of the disease disparities, as differential genetic and epigenetic alterations have been reported to account for higher PCa incidences and mortality rates in AA men compared to EA men [5,6]. Furthermore, a complex combination of socioeconomic factors such as education, income, and access to healthcare as well as lifestyle/environmental factors including tobacco smoking, binge drinking, and high meat intake may contribute to PCa and the disease’s disparities [5,6,7]. Scientific reports have established that PCa initiation and progression is dependent on androgens and androgens signaling via androgen receptors (ARs); therefore, depriving PCa cells of androgen and androgen receptor functions by inhibiting androgen biosynthesis or AR function will act to suppress tumorigenesis [8]. Consequently, therapies that target the AR function remain the cornerstone for treating men with advanced PCa, and various AR antagonists have been developed [8]. However, despite successful treatment with androgen deprivation therapy (ADT), some patients with PCa will progress to a lethal and incurable form of PCa, termed castration-resistant prostate cancer (CRPC) [9]. Clearly, there is an urgent need for alternative therapeutic approaches and treatment options to reduce PCa mortality rates. Surprisingly, phytochemicals and carotenoid extracts from certain dietary vegetables, teas, herbs, medical plants, and spices such as saffron have been shown to suppress experimental carcinogenesis in various organs in pre-clinical models [10]. Saffron (*Crocus sativus* Linnaeus) is widely used in cooking as a spice and flavoring agent and is also known for its medicinal and pharmacological properties. There is a growing number of empirical studies that demonstrate the effectiveness of saffron products in fighting cancers. Saffron is principally composed of crocin, picrocrocin, crocetin, and saffranal [11].

Several reviews have reported the anti-inflammatory effects of saffron treatment in gastrointestinal disorders and saffron’s antitumor effects on gastric cancer, colorectal cancer, liver cancer, and pancreatic cancer [12,13]. In prostate cancer, preclinical studies of the oral treatment of saffron and its major metabolites crocetin (CCT) and crocin (CR) in aggressive-PCa-cell-line xenografts in male nude mice also showed antitumor effects [14]. It was shown that in PCa-cell-line-derived xenografts, saffron and its main metabolites CR and CCT inhibited PCa cell invasion and migration and reverted the epithelial-mesenchymal transition (EMT), as evidenced by the significant reduction in N-cadherin and beta-catenin expression and the increased expression of E-cadherin [14]. One other report showed that saffron and its main metabolite crocin inhibited cell proliferation, caused cell cycle arrest, and induced apoptosis via caspase-dependent pathways in a panel of non-malignant and malignant PCa cell lines [15]. The accumulating scientific data of the antitumor and chemo-preventive properties of saffron and its main metabolites indicate that mechanistically, saffron modulates cellular growth and progression; inhibits cancer cell proliferation and antioxidant activity; and regulates immune signals, apoptosis, and the inhibition of DNA and RNA synthesis as well as free-radical chain reactions [16].

Given the potential chemo-preventive and chemotherapeutic properties of saffron in PCa, there is a need to fully elucidate the molecular mechanisms of saffron’s properties. In the present study, we investigate the effect of saffron on epigenetic mechanisms, DNA repair, and inflammatory pathways in in vitro PCa cell lines.

## 2. Materials and Methods

### 2.1. Prostate Cancer Cell Culture

The European American PCa cell lines PC3 and LNCaP were purchased from the American Type Culture Collection (ATCC, Manassas, VA, USA). The cell lines were maintained in RPMI-1640 supplemented with 10% fetal bovine serum (Gibco/Thermo Fisher, Waltham, MA, USA), 100 ug/mL streptomycin, and 100 U/mL Penicillin (Cellgro, Manassa, VA, USA). The African American prostate cancer cell line MDA-PCa-2b was obtained from ATCC and cultured in F-12K medium (Corning, Glendale, AZ, USA) supplemented with 20% fetal bovine serum, 1% Penicillin/Streptomycin (Corning, Glendale, AZ, USA), 25 ng/mL cholera toxin (Sigma-Aldrich, St. Louis, MO, USA), 10 ng/mL human epidermal growth factor (Sigma-Aldrich), 5 μg/mL insulin (Sigma-Aldrich), 100 pg/mL hydrocortisone (Sigma-Aldrich), 5.8 ng/mL selenous acid (Sigma-Aldrich), and 700 ng/mL O-phosphorylethanolamine (Sigma-Aldrich). All cell lines were cultured in a humidified 5% CO_2_ air atmosphere at 37 °C.

### 2.2. Saffron Treatment and Cell Viability Assay

Experiments using powdered crude saffron extracts (Gulf Pearls SPRL Brussels, Belgium, www.gp-food.com, accessed on 15 February 2023) were carried out as previously described [17]. Briefly, saffron was dissolved in ultra-pure water at a final concentration of 20 mg/mL and mixed on an orbital shaker in the dark for 1 h before being used at different concentrations. The composition of saffron crude extracts was 5 mM safranal, 5 mM crocin, and 5 µM crocetin [17].

All cells were seeded at a density of 50,000 cells/well in 12-well plates in 1 mL of complete growth medium. Cells were allowed to attach for 24 h before being treated with different concentrations of saffron extracts for different time points. LNCaP and DU145 cells were plated in a 24-well plate and treated with water-soluble saffron in a time- and dose-dependent manner (0 mg/mL, 0.5 mg/mL, 2 mg/mL, and 4 mg/mL) with 1 mL of media. At 24, 48, and 72 h post-treatment, cells were trypsinized and counted using a Bio-Rad TC20 automated cell counter. The effect of saffron crude extracts on LNCaP cell viability after treatment with water-soluble saffron in a time- and dose-dependent manner (0 mg/mL, 0.5 mg/mL, 2 mg/mL, and 4 mg/mL) was determined using a direct cell counter after trypan blue staining. The experiments were carried out in triplicate.

### 2.3. RNA Extraction and Real-Time and Standard PCR

Total RNA was isolated from treated and untreated cultured cells using TRIzol reagent (Invitrogen, Carlsbad, CA, USA) according to the instructions of the manufacturer. Total RNA (2 μg) extracted from cultured cells (70% confluence) was reverse-transcribed to cDNA with the iScript cDNA synthesis kit (Bio-Rad, Hercules, CA, USA) following the protocol of the manufacturer. Real-time quantitative RT-PCR was carried out for different gene amplicons using Taqman assays (Appendix A) in a CFX96 real-time PCR machine (Bio-Rad) in 35 cycles of 95 °C for 5 s, 60 °C for 30 s, and 72 °C for 60 s. The C_t_ values in the log linear range—representing the detection threshold values—were expressed in ΔC_t_ (C_t_ value of target gene minus C_t_ value of housekeeping β-actin gene). Primers sequence information for individual genes is as reported in Appendix A. The β-actin Taqman assay or standard PCR primers were used as endogenous control, and RT-PCR analysis was performed in triplicate.

### 2.4. Morphologic Analysis Using an Inverted Microscope

Morphological studies using a normal inverted microscope were carried out to observe the morphological changes of cell death in LNCaP cell lines in response to saffron extract treatment (4 mg/mL). The untreated cells served as the negative control. The morphological changes of the cells were visualized under the normal inverted microscope at Day O (DO), D1, and D2 post-treatment.

### 2.5. Western Blotting

Cells were harvested and lysed in a radioimmune precipitation assay (RIPA) buffer, 0.5 M EDTA, and protease and phosphatase inhibitors cocktail. Total proteins were quantified using a BCA protein assay kit (Pierce Inc, Rockford, IL, USA). For Western blots, 30 mg of protein extract/lane was electrophoresed, transferred to nitrocellulose membrane (Invitrogen), and incubated overnight with each of the following primary mouse monoclonal antibodies: C-Myc (1:500; sc-40; Lot#J0220), NF-kB (1:200; sc-8008; Lot# H1220), AR (1:100; sc-7305; Lot#J2920), HDAC1 (1:200; sc-81598; Lot# J0822), PTEN (1:200; sc-7974; Lot# 10420), TNFα (1:500; sc-515765; Lot# 11020), N-Ras (1:200; sc-31; Lot# J0520), Caspase-3 (1:200; sc-7272; Lot# J0422), HDAC3 (1:200; sc-376957; Lot#G0323), HCAM (1:200; sc-7297; Lot# D2123), MBD2 (1:200; sc-514062; Lot# B0922), Dnmt1 (1:200; sc-271729; Lot# A1323), and Dnmt3b (1:200; sc-393845; Lot# K2316). The GAPDH antibody (1:5000; sc-32233; Lot# J2020) was used as an internal loading control. Membranes were washed and incubated with anti-mouse secondary antibody (1:2500; sc-2005). All antibodies were purchased from Santa Cruz Biotechnology (Dallas, TX, USA). The antigen–antibody reactions were visualized using an enhanced chemiluminescence (ECL) assay (Bio-Rad; Hercules, CA, USA), and the membrane was imaged using a digital imager/ChemiDoc MP imaging system (Bio-Rad). A densitometry program using ImageJ version 1.54 (https://imagej.nih.gov/ij/, accessed 18 October 2023) was used to quantify bands in the Western blot, and the protein expression level was displayed as the ratio of each protein to the GAPDH protein level. The data are representative of triplicate experiments.

### 2.6. Statistical Analysis

All experiments were repeated three times, and the results are presented as the mean ± SD. Analyses of significance were performed using the Student’s *t*-test, Fisher test, or one-way ANOVA. *p* < 0.05 was considered statistically significant.

## 3. Results

To assess the effect of saffron extract on PCa proliferation, we incubated the PCa cell lines LNCaP, DU145, and MBA PCa 2b with different concentrations of saffron extract dissolved in water, as previous reports indicate that water-solubilized saffron and its major metabolites have potent antitumor effects in PCa cell models [14]. We diluted the appropriate volume of the saffron extract solution in complete growth medium to obtain the desired concentration. To assay for the cytotoxic activity of 4 mg/mL of saffron extract, 200 µL of the saffron extract (20 mg/mL) was added to 800 µL of the growth medium for 50,000 cells. Controls were carried out by adding 200 µL of water to 800 µL of complete growth medium for 50,000 cells. Cell proliferation was evaluated at 24, 48, and 72 h time points using a Bio-Rad TC10 automated cell counter. Results (Figure 1), expressed as the % proliferation of different drug concentrations relative to untreated control cells, demonstrated that at the highest concentration of saffron treatment (4 mg/mL), there was more than a 50% growth inhibitory effect on LNCaP and MBA PCa 2b cells, whereas modest inhibition was observed in the androgen-independent DU145 cells at the optimum 48 h inhibition time point. Results indicate that saffron inhibits cell proliferation in a dose dependent manner, whereby significant inhibition of cell proliferation is found at a 4 mg/mL concentration compared to the untreated control cells. Our results are consistent with previous analysis whereby saffron extract and its major metabolite crocin demonstrated anti-proliferative effects on malignant prostate cancer cell lines in a time- and dose-dependent manner, with IC50 values ranging between 0.4 mg/mL and 4 mg/mL [18,19]. To test the cytotoxicity of varying concentrations of saffron extracts on PCa cell viability, LNCaP cells were treated with saffron extracts at different concentrations (0.0 mg/mL, 0.5 mg/mL, 2 mg/mL, and 4 mg/mL) for 24, 48, and 72 h (Appendix A). The results of saffron treatment showed a dose- and time-dependent inhibition of cell viability in LNCaP cells (0% to 71%).

Next, we investigated the morphological changes of LNCaP cells in response to treatment with 4 mg/mL of saffron extract at 0, 24, and 48 h post-treatment. While the MDA PCa 2b and LNCaP cell lines showed significant growth inhibition in response to saffron treatment (Figure 1), we noticed that MDA PCa 2b cells grew by clustering together and showed more active proliferation than LNCaP cells; therefore, MDA PCa 2b cells are not ideal for monitoring cell morphology (results not shown). This observation also indicates that the treatment was not toxic to the MDA PCa 2b cells. Figure 2A shows that after 24 h of treatment, morphological changes were observed in LNCaP cells with respect to the untreated LNCaP control cells; this consisted of a reduction in the number of living cells. After 48 h of saffron treatment, cytotoxic effects were prominent. Nearly all cells were granulated, cell proliferation was inhibited, the cells were fragmented, the forms of the cells were spherule, and cellular detachment was significant. The anti-apoptotic oncoprotein Bcl-2 is known to inhibit apoptosis induced by a variety of physiological and pathological stimuli, and it can be used to measure levels of cell death by apoptosis in tissue. We tested the mRNA expression of Bcl-2 by PCR. In addition, we tested the inflammatory response of IL-2 in response to saffron treatment. We observed a decreased expression of Bcl-2 and increased expression of IL-2 in response to saffron treatment (Figure 2B), suggesting that the treatment induces cell apoptosis. Broadly, the increase in IL-2 expression might be due to the induction of an immune response as well as the increased production of macrophages and natural killer (NK) cells, which are characteristic of programmed cell death and the modulation of the inflammatory response and antitumor activity [20].

To ascertain the molecular mechanisms underlying saffron’s antitumor effects in prostate cancer cells, we performed a qRT-PCR analysis of several genes involved in DNA repair and epigenetic signals in response to saffron treatment in prostate cancer LNCaP cells. We examined the gene expression of several epigenetic regulators in LNCaP cells treated with saffron compared with the controls (Figure 3A). We observed that saffron significantly reduced the expression patterns of catechol-O-methyltransferase (COMT) 0-6-methylguanine-DNA methyltransferase (MGMT) and euchromatic histone lysine methyltransferase 2 (EHMT2) as well as NAD-dependent protein deacetylase sirtuin 1 (SIRT1) transcripts in the saffron-treated cells compared with the untreated controls for both D1 and D2 time points. We also investigated the expression of several genes involved in DNA repair pathways in response to saffron treatment (Figure 3B). We observed a significant reduction in the expression of RecQ-like helicase (WRN) and WD repeat domain 70 (WDR70) in saffron-treated LNCaP cells at both D1 and D2. On the other hand, we observed more than 2-fold upregulation of p53, RecQ-like helicase 5 (RecQ5), and macrophage-stimulating 1 receptor (MST1R) in saffron-treated LNCaP cells at D1; however, the expression levels of p53, RecQ5, and MST1R were dramatically reduced at D2 in the saffron-treated cells compared with the untreated controls. These studies were performed twice, and results represent the average of both analyses. All data at the mRNA level were normalized based on β-actin expression in the corresponding groups.

Finally, we were interested to assess the effect of saffron on epigenetic regulators in prostate cancer. In silico RNA sequencing of prostate adenocarcinoma (PRAD) TCGA analysis demonstrated upregulation of DNMT3B, DNMT1, and MBD2 in tumors (*n* = 497) compared to normal (*n* = 52) samples (Figure 4A–C), suggesting a global increase in DNA methylation, which might be associated with gene-specific downregulation [21,22]. We carried out Western blot analysis to detect the expression of the several key epigenetic regulatory genes DNMT3b, MBD2, and DNMT1 as well as other key genes involved in prostate carcinogenesis (CD44, Caspase 3, NF-kB, TNFα, AR, N-RAS, C-myc, PTEN, and GAPDH (housekeeping internal control)) in saffron-treated LNCaP cells compared to untreated controls (Figure 4D). Densitometry results of Western blot analysis showed a decrease in DNMT3B, DNMT1, and MBD2 in prostate cancer cells upon treatment with saffron compared to untreated cells (Figure 4E), suggesting a global decrease in DNA methylation which might be associated with increased specific genes in prostate cells (increased tumor-specific gene expression, due to decreased methylase levels). In addition, we observed a decreased expression of CD44, AR, N-RAS, TNFα, NF-kB, and C-myc in saffron-treated LNCaP cells compared with the untreated control. Furthermore, we observed an increased expression of AR and PTEN in the saffron-treated cells, compared with the untreated control cells. Our data support previous observations that saffron plays a key role in modulating tumor-specific gene expression either directly or indirectly, which indicates that saffron might exert a survival effect via diverse signaling pathways that are important in carcinogenesis, as reviewed by Boozari and Hosseinzadeh [23].

## 4. Discussion

Saffron is a natural spice widely used as a food flavoring and a medical plant for thousands of years. Several studies have documented the health benefits of saffron and its major metabolites for their antitumor, antioxidant, anti-inflammatory, and immunomodulatory effects with regard to the prevention and treatment of several cancers, including lung, liver, breast, stomach, colorectum, cervix, and prostate cancers [15]. Some of the mechanistic actions attributed to the antitumor properties of saffron in many cancer cells include modulation of the carcinogen metabolism, regulation of cell growth and cell cycle progression, inhibition of cell proliferation, enhancement of cell differentiation, stimulation of cell-to-cell gap junction communication, apoptosis, and retinoid-dependent signaling [16]. Other studies have shown that saffron’s antitumor properties are attributed to its bioactive compounds, crocin and crocetin [24].

In the present study, we investigated the antitumor properties of saffron’s main metabolites: crocin, safranal, and crocetin. These were extracted by grinding saffron (stigma from *Crocus sativus*) in liquid nitrogen [25]. We assessed the different dose responses of saffron extract (0 to 4 mg/mL) on PCa cells in vitro and observed significant antitumor effects of saffron at 4 mg/mL concentrations in PCa cells. A limitation to our in vitro findings of PCa cell responses to 4 mg/mL saffron is that they cannot be extrapolated to in vivo human conditions. This is because the amount used in this study may not be biologically relevant for in vivo settings, because if such a small amount of saffron is ingested in humans, only a portion of the active metabolites may cross the gut wall. In addition, these metabolites will be diluted in the blood and lymph volumes and hence the resulting concentrations will be very low (<1 µg/mL) [26]. Furthermore, most of the compounds will be metabolized before they can reach the PCa cells, and some of these metabolites will be inactive. In prostate cancer, all the scientific evidence for saffron’s antitumor effects has been obtained from preclinical in vitro and in vivo studies in PCa cell lines and mouse xenograft models; therefore, there is a need for clinical trials to ascertain the dosage of saffron extract in human PCa through a series of pharmacological, pharmacokinetic, and toxicological studies. We are currently conducting a clinical trial that stems from a successful in vitro and in vivo data extrapolation to humans with ulcerative colitis [12,17]. In this clinical trial, we used two saffron concentrations after extrapolating from mice findings to human body weight [13]. We are assessing the presence of saffron compounds in the plasma of the recruited patients. A critically observed local effect of saffron metabolites on colon cancer cells is a systemic effect on the immune signature (switched from pro to anti-inflammatory). Our present study is still in its early stages, but we do see a positive effect of saffron extract on PCa cell lines. Our next steps will be to determine effective concentrations of saffron extract on prostate cancer in mouse models before extrapolating to humans. However, we are highly optimistic that we will be able to start a clinical trial on prostate cancer patients very soon in the foreseeable future.

Clinical studies of saffron extracts have been conducted in other disease settings. One study that investigated the metabolism of saffron in humans was performed using a sailuotong (SLT) capsule to ascertain the clinical and pharmacological relevance of SLT. Sailuotong capsules are a standard herbal medicine formulated from a mixture of ginseng (the dried root and rhizome of Panax ginseng C. A. Meyer), ginkgo (the leaves of *Ginkgo biloba* L.), and saffron (the stigma of *Crocus sativus* L.) extracts; they are proven to have health benefits for treating vascular dementia [27]. In this study, two healthy male volunteers took eight capsules of SLT (480 mg) with 250 mL water after fasting overnight, and venous blood (3 mL) was collected before oral administration of the capsules and at different time points until 72 h post-treatment. Seventeen metabolites were analyzed in the blood, and the saffron metabolite crocetin was detected in the blood until 24 h post-administration and at a concentration of 100 ng/mL, though with large individual differences. Another cohort of 10 healthy volunteers who had fasted for 12 h were given 300 mg of saffron extract. Blood samples were collected before and after ingestion of the saffron extracts, and the absorption profile of crocetins was monitored in blood serum for 5 h [28]. The study showed a maximum concentration of crocetin in blood serum at 90 min post-ingestion and demonstrated that circulating saffron metabolites were neuroprotective in ex vivo clinical settings. None of the volunteers who received a single dose of 300 mg of saffron extract reported any adverse effects. Indeed, the main reported side effects of saffron overconsumption were related to bleeding and vascular dysfunction. However, two other clinical studies that investigated the potential effect of saffron on blood coagulation and platelet aggregation showed that the administration of saffron extracts at 200 and 400 mg/day for 7 days exhibited no adverse side effects [29,30]. At the higher dose of 400 mg/day, saffron induced changes in the hematological parameters, including decreased red blood cells, hemoglobin, and platelets. However, these alterations were within the normal range and therefore not clinically important. Another pharmacokinetic study by Almodovar et al. [26] on the administration of two different concentrations of saffron extract tablets (56 mg and 84 mg) showed a maximum crocetin blood concentration between 60 min and 90 min post-ingestion, suggesting that the kinetics may depend on the dose of saffron extract.

The accumulating scientific evidence demonstrates that crocin and crocetin inhibit tumor growth in several cancers—including colorectal, pancreatic, breast, and prostate—as well as chronic myelogenous and leukemia. Crocin metabolite has been extensively investigated and shown to inhibit telomerase activity, microtubule polymerization, cyclin D1, nuclear factor kappa B (NF-kB), multi-drug-resistance-associated protein (MRP1), and MRP2 overexpression. The crocin metabolite also induces apoptosis through multiple pathways, including the activation of caspase 8, increased p53 expression and Bax/Bcl-2 ratio, and down-regulated survivin and cyclin D1 [31]. In addition, crocin can inhibit tumor invasion and metastasis through down-regulation of matrix metalloproteinase 2 and 9 (MMP2 and MMP9), N-cadherin, and beta-catenin expression and cell cycle suppression at G1, G0/G1, S, and G2/M phases [31]. Crocin has also been reported to induce antiproliferation in a dose-dependent manner in both hormone-sensitive and hormone-insensitive prostate cancer cell lines [15]. Another report found that crocin suppresses the re-proliferation of quiescent prostate cancer cells in vitro via down-regulation of important regulatory transcription factors such as Skp2, E2F1, NF-kB, C-myc, and other cell-cycle-regulatory genes [32].

In our study, saffron showed stronger antiproliferative effects in the androgen-sensitive PCa cells compared to the androgen-insensitive PCa cell lines. However, we observed increased expression of AR in saffron-treated LNCaP cells, suggesting that the anti-proliferative effect of saffron on LNCaP cells may depend on AR levels, as previously shown for the chemo-preventive effect of vitamin D on LNCaP cells [33]. Furthermore, we observed up-regulation of PTEN in saffron-treated LNCaP cells, as previously observed in LNCaP treated with curcumin [34]. Thus, saffron can be added to the growing list of natural compounds with potential anti-cancer mechanisms and therapeutic potentials in the context of prostate cancer prevention and treatment [35].

Several reports have indicated that saffron antitumor effects may be mediated through interaction with the DNA causing epigenetic changes. Ashrafi et al. [36] used spectrophotometry and spectro-fluorometry analysis to demonstrate that one possible mechanism of saffron antitumor action is via the reduced interaction of histone H1 with DNA molecules in the presence of saffron carotenoids. Given that H1 plays a direct role in stabilizing the nucleosomes and higher-order structure of chromatin and functions as a general or specific repressor of transcription by limiting the access of transcription activators to chromatin, saffron’s effect of inhibiting HI-DNA binding interactions or depleting H1 may be a necessary step in the activation of many genes. Aberrant epigenetic modifications are hallmarks of PCa, and previous observations indicate that saffron carotenoids can directly or indirectly regulate epigenetic changes and alter gene expression profiles. Thus, we were interested to investigate the expression of several epigenetic factors in PCa cell lines that were treated with saffron extract or left untreated, using Western blotting. We have also shown that saffron decreased the expression of the epigenetic enzymes SIRT1, EHMT2, MGMT, and COMT at the mRNA level and DNMT1, DNMT3b, and HDAC3 at the protein level in LNCaP cells. Extrinsic and intrinsic environmental factors such as pollutants, hormones, and active dietary components can impact epigenetic mechanisms and contribute to cancer development. For instance, sirtuin 1 is a nicotinamide adenine dinucleotide (NAD)+- dependent protein deacetylase that acts as a sensor that directly connects metabolic perturbations with transcription outputs. Various reports have shown that SIRT1 is not just a histone deacetylase: it also interacts and regulates the activity of many co-regulators and transcriptional factors such as FOXO [37]. Other studies have shown that saffron extracts decrease HDAC1 and HDAC3 expression in glioblastoma cell lines under treatment with saffron extracts in a dose-dependent manner [38]. Thus, one antitumor property of saffron arises through the regulation of epigenetic-related process genes.

The nuclear factor-kb (NF-kB) transcription factor has important roles in many eukaryotic cellular processes, including angiogenesis, inflammation, cell proliferation, transformation, and tumorigenesis. NF-kB is primarily involved in the regulation of the human necrosis factor (TNFα) signaling pathway. Our observation of decreased expression of NF-kB and TNFα is consistent with saffron’s antitumor properties, as reported in other cell systems [39,40].

Apoptosis is an important mechanism that contributes to cell-growth reduction and is reported to be induced by saffron extract in different cell types [24]. Treatment of the androgen-dependent prostate cancer cell line LNCaP with saffron induced decreased expression of CD44 and Bcl-2; these are hallmarks of apoptosis [38,41], which clearly supports saffron’s anti-tumor function via apoptosis.

We report an increased expression of DNA repair genes WRN, RECQ5, and WDR70 and the tumor suppressor genes p53 and MSTR1 in response to saffron treatment for 24 h. However, there was a dramatic decrease in the expression of these genes after 48 h of saffron treatment. The difference in expression suggests a biphasic time-dependent and perhaps dose-dependent effect of saffron treatment. As previously reported, saffron seems to be a biphasic, dose-dependent substance; at a certain dose, it has direct anti-cancer properties; at other doses, it acts as a scavenger against ROS. Both roles are anti-tumorigenic. These properties make saffron a very interesting substance to pursue as a therapeutic anti-cancer agent [42,43]. Thus, at least for the DNA repair genes, a longer exposure to saffron is necessary to inhibit their expression and block DNA repair in LNCaP cells. Saffron’s anti-proliferative effects have been previously reported to be significant in colon cancer cells with a deficiency in mismatch repair genes [44], indicating another potential target for saffron’s mechanism, via DNA repair in prostate-cancer-proficient MMR cell lines.

## 5. Conclusions

Our observation of the antitumor effects of saffron extracts in prostate cancer cells supports previous studies of the multiple molecular pathways that saffron metabolites use to exert biological effects upon cancer. Given the potential health benefits of saffron extracts, more studies are needed to fully elucidate the antitumor properties of saffron in prostate carcinogenesis.

## Figures and Tables

**Figure 1 nutrients-16-00114-f001:**
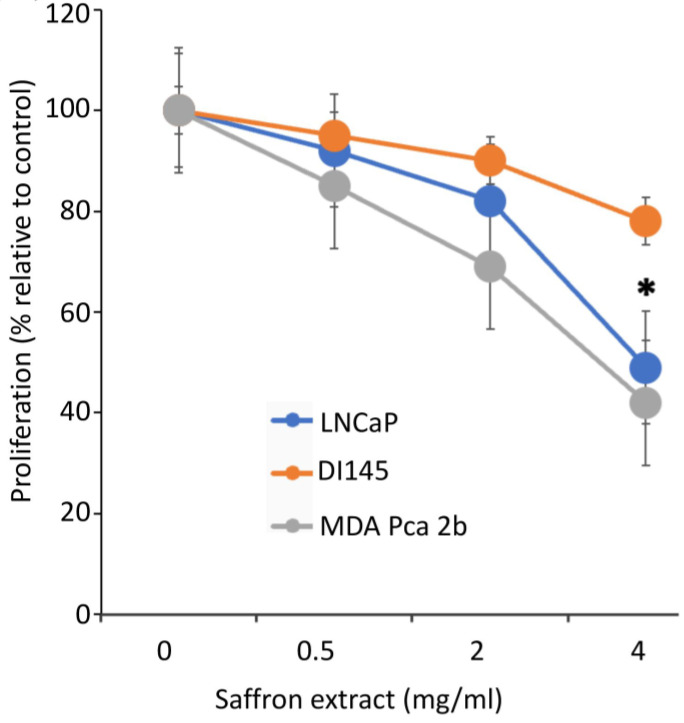
Cell proliferation assay. Dose-dependent effects of aqueous extracts of saffron measured after 48 h of treatment in androgen-dependent (LNCaP and MDA PCa 2B) cell lines and androgen-independent (DU145) cell lines. * (*p* < 0.05) shows statistically significant inhibition in both the LNCaP and MDA PCa 2b cell lines.

**Figure 2 nutrients-16-00114-f002:**
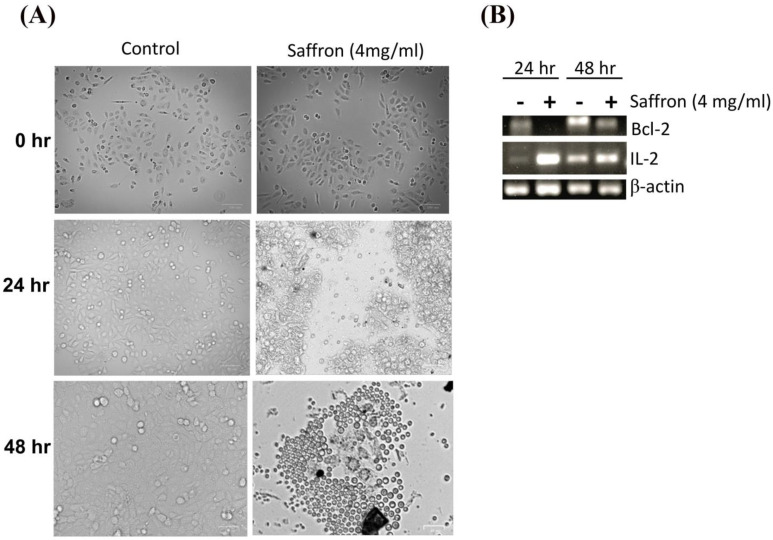
(**A**) Cell morphology changes (as measured by an inverted microscope) in response to saffron treatment (4 mg/mL) or control (untreated cells) for 24 h and 48 h time points. (**B**) PCR analysis of Bcl-2 and IL-2 expression and β-actin (internal control) for saffron treatment (+) and control groups (-; untreated cells) for 24 h and 48 h.

**Figure 3 nutrients-16-00114-f003:**
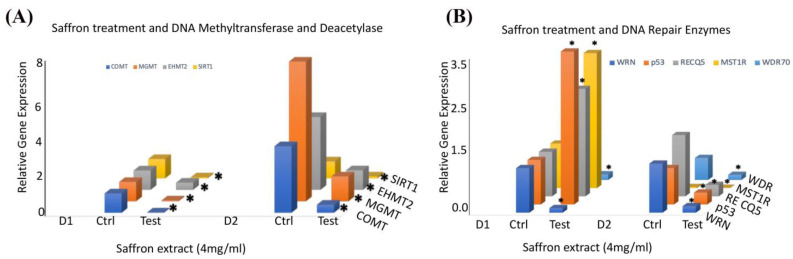
(**A**) The relative mRNA transcript expression levels of COMT, MGMT, EHMT, and SIRT1, as analyzed in LNCaP cells treated with saffron (4 mg/mL) or untreated (Ctrl) at D1 (24 h) and D2 (48 h) using RT-PCR and expressed relative to β-actin to correct for variations in the amount of reverse-transcribed RNA. (**B**) The relative mRNA transcript expression levels of WRN, p53, RECQ5, MST1R, and WDR70, as analyzed in LNCaP cells treated with saffron (4 mg/mL) or untreated (Ctrl) at D1 (24 h) and D2 (48 h) using RT-PCR and expressed relative to β-actin to correct for variations in the amount of reverse-transcribed RNA. * (*p* < 0.05) shows statistically significant changes in the control (Ctrl) versus saffron treatment groups.

**Figure 4 nutrients-16-00114-f004:**
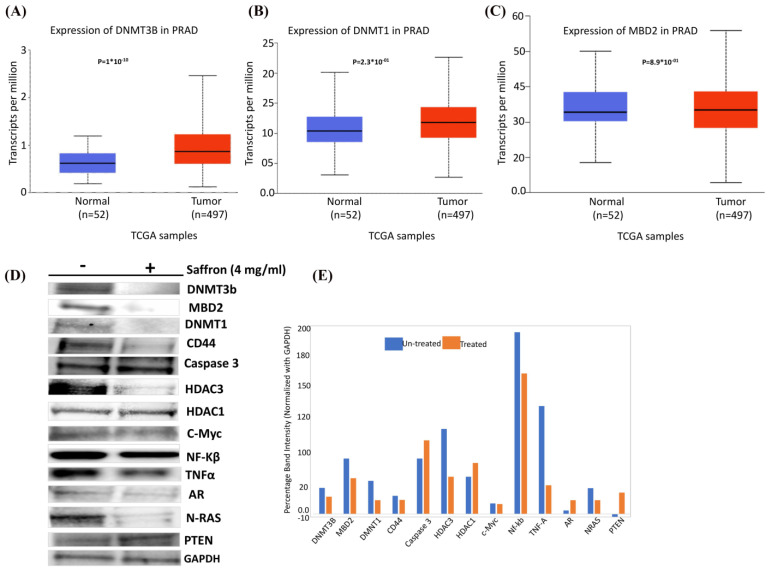
In silico RNA sequencing of TCGA-PRAD and Western blot analysis. (**A**) Expression of DNMT3B in TCGA-PRAD. (**B**) Expression of DNMT1 in TCGA-PRAD. (**C**) Expression of MBD2 in TCGA-PRAD. (**D**) Protein extracts collected from LNCaP cells treated with saffron (4 mg/mL) or untreated control (-) for 48 h and analyzed by Western blotting with the following antibodies: DNMT3b, MBD2, DNMT1, CD44, Caspase 3, HDAC3, HDAC1, C-Myc, NF-kB, TNFα, AR, N-RAS, PTEN, and GAPDH. Data represent 3–4 independent experiments. (**E**) Densitometry of Western blot analysis, shown as the ratio of each protein expression to GAPDH protein. Statistical significance is indicated (*p* < 0.05; *t*-test).

## Data Availability

The data presented in this study are available on request from the corresponding author. The data are not publicly available due to privacy.

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
