# Peer review of "Mechanism of Antitumor Effects of Saffron in Human Prostate Cancer Cells"

_nutrients, 2023, doi:10.3390/nu16010114_

Round 1

Reviewer 1 Report

Comments and Suggestions for Authors

The article is entitled: Mechanism of Antitumor Effects of Saffron in Human Prostate Cancer Cells and is proposed by Khan M, Hearn , Parry C, Rasid M, Brim H, Ashktorab H, and Kwabi-Addo B.

General remarks

This manuscript may be published according to the editor however major revisions are required before acceptance.

Although the approach used in this study is rather classic it includes many bias that should be highlighted in the discussion part and also at least evoked in the abstract.

The authors used a crude extract of saffron regardless to its precise composition especially in the known active compounds i.e. crocin and crocetin. This precise composition should be added in the material and method part. Saffron being also known to contain picrocrocin, safranalor, kaempferol, quercetin and isorhamnetin the known action of these compounds are never discussed nor their presence in the extract.

In addition, although the tests performed in this study are common, the authors and the readers should be aware of the limitations of these kinds of approaches. Indeed, in vivo in human, only a small amount of saffron should be ingested and only a part of the active compounds may cross the gut wall. They will be diluted in the blood and lymph volume (>15 litres) for the less and therefore the resulting concentrations will be very low (<1 µg/ml according to DOI: 10.1155/2020/1575730). Most of the compounds will be metabolised before reaching the PCA cells and some of these metabolites will be inactive. Therefore, before concluding to an effect of saffron extract on PCA cells, the authors should emphasize the clinical effects of this plant on prostate, if any has already been documented. If such data exist they should be mentioned in the abstract because they fully justify the present work. Indeed, in vitro data are usually produced to sustain a clinical effect and explain its mechanism. They rarely prove to be reliable per se.

I would suggest the author to add a paragraph in the discussion dealing with the plausibility of the in vivo effects of the saffron extract in humans. This part should contain data on what is known about the composition of a saffron extract (DOI: 10.1007/s11130-020-00873-5), and although on the extract used in this study. A metabolomic analysis would be very interesting. The paragraph should also talk about the metabolism of the active compounds in vivo preferentially in humans (DOI: 10.1016/j.jep.2022.115843) and mention a study that involved human plasma serum extract obtained after ingestion of Saffron and its in vitro effect (DOI: 10.3390/nu14071511). Finally it should evoke the pharmacokinetics of these compounds still in human (DOI: 10.1155/2020/1575730). Of course if clinical data already exist showing undoubtedly that saffron have health effects in human especially related to prostate, they should be included in this paragraph.

Materials and Methods

The concentrations of extracts that were used in this study are huge compared to the Cmax reported in (DOI: 10.1155/2020/1575730) (<1µ g/ml). Indeed, when xenobiotics are considered it should be kept in mind that their action, if any, can pass via many different pathways and unfortunately the pathways which are used are not the same according to the dose. This is why it is important to test doses that are relevant to the in vivo situation. The authors should comment on their choice. Did they test lower concentrations and if yes what did they observe. Are there data on plasma concentrations of crocin, crocetin and quercetin in human beings? The viability test is not mentioned in the cell treatment. Indeed, looking at the huge doses of extracts used directly on the cells there is a serious risk of cell death.

Line 137, Day 0 (Zero) and not Day O.

Results on cell cultures

I am not fully convinced that the authors measured a proliferation %. Indeed, proliferation means that the cell number increases in the plate-wells. Here the only thing which is mentioned is a percentage of viable cells that are present in the tested wells, in reference to the control treatment. However the comparison of the growth kinetic of the cells during the 48h of treatments is missing. Can the author add this figure to their manuscript? It can be proposed as a supplementary material.

Figure 1. is more a kinetic of cell numbers according to cell lines and doses of saffron extract. Because the mother solution of the crude saffron extract is at a 20mg/ml concentration, when the 4 mg/ml concentration is tested it means that the extract represents ¼ of the cell medium. Therefore, the cells have only ¾ of the full growth medium to live. Indeed, such condition are not optimal for cells. It would have been better to dissolve the extract in an organic solvent like ethanol or DMSO and thus at a much higher concentration. Then it could have been possible to add to the cells’ wells the same amount of solvent including in the control but this amount would have been small compared to the growth medium. From this it is unclear if it is the saffron effect which is measured or if it is a decrease in growth medium quality. Beware, DU145 is not written correctly. Indeed, there is a control missing, if the extract has been added in water and if the extract represent up to ¼ of the final cell medium, the right control would be the growth medium to which would be added water in the same proportion. Again, what appears from the treatment is the distress of the cells. For me the culture conditions are at least partly responsible for this distress and not specifically the saffron extract.

Figure 2. tends to show that the treated cells agglomerate around the extract matter. It is not mentioned what is the solubility of the extract in water and if it is fully dissolved at the doses used. Why did the authors only choose to present the cells under the 4 mg/ml treatment? Why did they choose to study only LNCaP cells? These choices should be explained. I think that the interpretation should have been more credible if the same results were shown on the three cell lines with an intensity reflecting their viability. What control could you propose to check for the global cells’ health?

Line 191. I would suggest saying “the treatment induce cell apoptosis”. I am not sure that this effect is specific to saffron. For me it could be also obtained using other extracts in the present conditions.

Discussion:

Line 260. The reference 18 is a paper dealing with in vitro tests. Again, such results cannot be used to indicate an in vivo effect in human. Can you please cite a work dealing with the clinical effect of saffron on men prostate cancer? Equally, the references 24 is an in vitro work and reference 16 is a sort of review.

Line 267-269. Please precise if these effects were shown in vitro or in vivo, un animals or in humans. To my knowledge, there is no clinical data on the effects of saffron on an established cancer. However, there are data that would indicate that saffron may be useful to cope with the chemotherapy engaged to cure cancers. Such effects are usually related to antioxidant molecules. Indeed there are antioxidant substances in saffron, like in onion for instance. Therefore, the authors should be cautious in their interpretations and conclusions.

Line 280. Please can you precise what are the range of doses that were tested in reference 18? They may be of the same order than the one used in the present study and therefore much higher than those ever found in human cells vicinity in vivo.

Line 292-296. Beware, the interpretation of in vitro results cannot be simply extrapolated to the in vivo situation especially when the doses used are pharmacological and when the results only show cell distress.

Please all over the discussion part precise the doses of pure compounds or extracts that have been used in previous studies.

Line 315. The reference 14 is not a clinical study but a study performed in mouse with huge doses of saffron (20 mg/kg/day). In addition it deals with a gut flora analysis while this flora is different between mice and humans.

Conclusions:

I would suggest to mention in the last paragraph the conditions of treatments: pharmacological doses much higher than the one previously measured in human plasma and in vitro results with variation of growth medium concentration. Therefore, although the results may be interesting per se, they cannot be transposed directly to the in vivo situation.

Comments on the Quality of English Language

There are some spelling mistakes sometimes to times.

Author Response

Response to Reviewer 1:

Thank you for the critical review of our manuscript entitled “Mechanism of Antitumor Effects of Saffron in Human Prostate Cancer Cells”

Abstract section:

The reviewer suggested the addition of a sentence to emphasize the clinical relevance of saffron on prostate.

In response we have included a sentence that in prostate cancer, dietary intake of saffron has been reported to have antitumor effect on aggressive prostate cancer cell lines derived xenografts in mice providing a rational for the current studies (line 20)

Material and Methods section:

The reviewer requested the addition of the precise composition of the saffron extracts in the materials and methods section:

In response we have included the composition and the final concentration of the saffron extract solution- The composition of saffron crude extracts was 5 mM safranal, 5 mM crocin and 5 μM crocetin (line 128-129)

We have also included the cell viability studies: The effect of saffron crude extracts on LNCaP cells viability after treatment with water soluble saffron in a time and dose dependent manner (0mg/ml, 0.5mg/ml, 2mg/ml, 4mg/ml) was determined using direct cell counter after trypan blue staining. The experiments were carried out in triplicates. (Line 136-139)

Data is presented as a supplement to figure 1.

Results section:

The reviewer asks for the rational for dissolving our saffron extract in water and not other solvents.

We include in the results section that saffron, and its major metabolites are highly soluble in water and shown to have antitumor effect in PCa cell models (line 188-189)

This Reviewer also questions the concentration range that we used in our studies. We include in the results of previous studies where the same concentration range has been studied. that our observation supports previous studies that showed that both saffron extract and its’ major metabolite crocin induced anti-proliferative effects on 5 different malignant prostate cancer cell lines in a time and concentration dependent manner with IC50 values ranging between 0.4mg/ml and 4mg/ml [23, 24]. (Line 197-201)

This reviewer also wants to know cell viability in response to the concentration saffron used in this study. We now present our cell viability analysis as supplementary figure 1 and also stated in the results- To text for the cytotoxic effect of saffron extracts on prostate cancer cell viability, LNCaP prostate cancer cells were treated with saffron extracts at different concentration (0.0 mg/ml, 0.5 mg/ml, 2 mg/ml, and 4 mg/ml) for 24-, 48- and 72-hours (supplementary Fig. 1). The results of saffron treatment showed dose and time dependent inhibition of cell viability in LNCaP cells (0% to 71%).

In figure 2- reviewer wants to know that rational for studying cell morphology only in LNCaP cells, we state in the results section that While both MDA PCa 2b and LNCaP cell lines showed significant growth inhibition in response to saffron treatment, the MDA PCa 2b growth by clustering together and show active proliferation than LNCaP cells for monitoring cell morphology (results not shown). Line 212-215

Discussion:

We appreciate the reviewer’s suggestions to ensure that references correspond appropriately to invitro and in vivo analysis.

We have re-written large portions of this manuscript ensuring that references are appropriate, have removed repetition and redundancies.

Per reviewer’s critical suggestions we now include a large paragraph in the discussion: line 309- 336.

-Describe how active saffron metabolites are extracted

-Discussing limitations of extrapolating our in vitro findings to in vivo human conditions

-Discussing studies that has been carried out on saffron metabolism.

-Clinical studies of saffron studies of saffron

-Pharmacokinetic studies of saffron extract

We are thankful to this reviewer for the critical review, we have carefully responded to the criticism and believe that this has substantially improved the quality of the manuscript.

Thanks you

Bernard Kwabi-Addo.

Reviewer 2 Report

Comments and Suggestions for Authors

This study was reported the utility of saffron as an anti-cancer agent for prostate cancer. The reviewer thinks that this paper seems very interesting. for readers. The reviewer would like to suggest some critiques as follows.

1.      Authors should check that the spaces are correct. i.e. line 47, line 51,…

2.      On line 54, please provide a citation for this sentence.

3.      In the Introduction, the contents discussing between Lines 37-68 and Lines 68-94 are so far different so that it is difficult to understand for readers.

4.      On line 67, what is “selective pressure of ADTs”? Is this correct?

5.      On line 68, the reviewer cannot understand from this sentence alone why the establishment of alternative therapies is so urgent.

6.      On line 83, is EMT correct? The epithelial-mesenchymal transition is correct?

7.      The reviewer thinks the method and the results are relatively clear.

Comments on the Quality of English Language

1.      Authors should check that the spaces are correct. i.e. line 47, line 51,…

Author Response

Response to Reviewer 2:

Thank you for the critical review of our manuscript entitled “Mechanism of Antitumor Effects of Saffron in Human Prostate Cancer Cells.”

Reviewer commented that in the introduction section, the contents discussing between lines 37-68 and lines 68-94 are so far different so that it is difficult to understand for readers:

In these introducing paragraphs, we started with the public health concerns of prostate cancer, discusses the biology, current treatment option and their challenges to set up the rational for novel therapeutic approaches using natural compounds.

On line 54 provide a citation- It is well known that PCa initiation and progression is dependent on androgen signaling pathway and for this reason depriving PCa cells of androgen and androgen receptor function through inhibition of androgen biosynthesis or AR function will act to suppress tumorigenesis [10]. Now line 62-65

What is the selective pressure of ADTs- we clarity this phrase in the introduction …… the result of selective pressure of ADTs whereby evolutionary force or mutational changes gives the tumor with a selective advantage in the absence of androgen [14]. Line 74-77

We have corrected the epithelial-mesenchymal transition (EMT) on line 93-94

Spacing and appropriate references have been checked or added.

We thank this reviewer for the fruitful review. We believe this has enhanced the quality of the manuscript.

Bernard. Kwabi-Addo.

Round 2

Reviewer 1 Report

Comments and Suggestions for Authors

I accept this new version but it would have been nice to discuss the dilution of growing cell medium by the extract tested at the highest concentrations. In addition, the effect detected on MDA PCA2B cells would indicate that the treatment is not toxic for the cells and this should have to be mentioned somewhere.

Comments on the Quality of English Language

No particular comment.

Author Response

We thank Reviewer this for his comments. In response to this reviewer's request, we now provide information discussing the dilution of growing cell medium by the extract tested at the highest concentration- this information is now provided in the results section (lines 174-179). 

We now mention that the saffron effect detected on MDA PCA2B morphology would indicate that the treatment is not toxic to the cells- this information is now provided in the results section (line 206-207). 

We thank the reviewer for his thorough review which we believe has enhanced the quality of this manuscript. 

Reviewer 2 Report

Comments and Suggestions for Authors

The authors well revised according to the reviewer’s recommendation.

Author Response

This reviewer is happy with our revision according to his/her recommendations. We thank this reviewer for the critical review as we believe this has enhanced the quality of our manuscript.